# Production of Liquid Biofuel Precursors: Optimization and Regulation of Lipase Fermentation and Its Application in Plant Oil Hydrolysis Process

Shuai Huang [1], Hao Li [1], Ruisong Zhu [2], Meng Wang [1,*] and Tianwei Tan [1]

[1] National Energy R&D Center for Biorefinery, Beijing University of Chemical Technology, Beijing 100029, China; 2019400024@mail.buct.edu.cn (S.H.)
[2] Petrochemical Research Institute, PetroChina, Beijing 102206, China
* Correspondence: wangmeng@mail.buct.edu.cn

**Abstract:** In the liquid biofuel production process, free fatty acids are important precursors for biodiesel and bio-aviation fuel. Enzymatic hydrolysis to produce free fatty acids has attracted more and more attention. However, enzymatic hydrolysis requires ensuring efficient expression of lipase and high product yield. In the shaking flask, the optimal addition amount of citric acid was 3 g/L, and the composition of composite amino acids was: histidine 0.1 mol/L, aspartic acid 0.03 mol/L and lysine 0.03 mol/L. On the basis of the shaking flask optimization, a 5 L fermentation tank was scaled up to obtain 25,000 U/mL of lipase after multiple batches of stable fermentation. This was the first time to cultivate *Y. lipolytica* using composite amino acid medium to produce high enzyme-activity lipase, demonstrating the industrial value of this lipase fermentation process. Finally, soybean oil was hydrolyzed to produce free fatty acids on a self-made static reactor. The optimized reaction conditions were: material flow rate 2813 mL/min, reaction temperature 45 °C, water content 25 wt% and lipase consumption 3 wt%. The yield of free fatty acids was 80.63% after 12 h under optimal reaction conditions in the self-made static reactor, 11.95% higher than that in the stirred reactor, which showed its industrial potential in the production of free fatty acids.

**Keywords:** liquid biofuel; precursors; enzymatic hydrolysis; 5 L fermentation tank; self-made static reactor

## 1. Introduction

Energy is the main driving force for a country's economic development and social progress. Currently, fossil fuels are still the world's main energy source, accounting for approximately 86% of the total energy output [1]. However, the continuous reduction of limited resources such as fossil fuels has raised awareness and concerns worldwide. In addition, utilization and combustion of fossil fuels pose environmental hazards and global warming [2,3]. Liquid biofuels are often considered one of the preferred sustainable combustion fuels to replace unsustainable fossil fuel energy sources [4–6]. Biodiesel mainly composed of fatty acid methyl ester (FAME), as a renewable and "carbon neutral" fuel of clean energy, is synthesized from various renewable raw materials, including various vegetable oils and animal fats. However, the process of synthesizing biodiesel from these raw materials with different levels of free fatty acids cannot be unified. The yield of biodiesel produced from soybean oil through transesterification was 96.4% at 40 °C when the reaction was performed for 3 h with catalyst loading of 5 wt%, molar ratio of alcohol to oil 15:1 and 20 wt% of cosolvent [7]. The natural bentonite-supported heterogeneous catalyst executed an excellent activity for waste cooking oil conversion as providing maximum biodiesel yield of 93% at optimal reaction conditions: reaction temperature 75 °C, molar ratio of alcohol to oil 10:1, catalyst amount 2.5 wt%, stirring rate 600 rpm and a reaction time of 3.5 h [8]. The maximum conversion rate of microalgal oil (*Spirulina platensis*) was 97.88% at 65 °C,

600 rpm, 2.5 wt% β-$Sr_2SiO_4$ catalyst, molar ratio of methanol to oil 12:1 and a reaction time of 104 min [9]. Due to the different free fatty acids of the raw materials, the catalytic process was not the same, and it was impossible to make an intuitive comparison between them. Therefore, raw materials with different acid values can be uniformly converted into liquid biofuel precursors—free fatty acids. Then, the unified raw material—free fatty acids—are converted to produce liquid biofuel.

At present, there are three main ways to produce free fatty acids: steam cracking, saponification and enzymatic hydrolysis [10,11]. The main technology for producing free fatty acids at the industrial tonnage level is the most well-known Colgate–Emery process [12]. Not only does this process require high-temperature and high-pressure reaction conditions (high temperature 250 °C, high pressure above 50 bar), but also many side reactions occur, such as the oxidation, dehydration and esterification reactions of triglycerides [10,13]. The saponification method which involves alkaline hydrolysis requires higher energy consumption than enzymatic hydrolysis, and there is also the formation of by-product soap in the intermediate step, which poses significant difficulties for product separation in the later stage [10]. Compared to the previous two methods, the enzymatic reaction conditions are mild and reduce the energy input. Due to the specificity of the enzyme reaction, there are relatively fewer by-products, making it easy to separate in the later stage. However, one of the biggest obstacles currently limiting enzymatic hydrolysis is finding a suitable and stable lipase that can achieve a good conversion rate [10].

Lipase (EC 3.1.1.3), also called triacylglycerol acylhydrolase, can hydrolyze carboxylate ester bonds and is an important member of the hydrolytic enzyme family [14]. Initially, Eijkman successfully isolated lipases from *Serratia marcescens*, *Pseudomonas aeruginosa* and *Pseudomonas fluorescens* [15]. To this day, it was widely recognized that various organisms including animals, plants and microorganisms could produce lipase. Unfortunately, lipase extracted from animal pancreas contained too many animal viruses and hormones and was easily broken down in the pancreas, making it unsuitable for use in the food industry. In order to obtain low-cost and stable lipases that were easy to synthesize, most research and industry used lipases from microbial sources [16]. The microorganisms that produced lipase were usually isolated from the environment where lipids existed. Common microorganisms included yeast *Yarrowia lipolytica* [17] and *Candida Antarctica* [18], as well as the bacterium *Pseudomonas aeruginosa* [19]. These microorganisms could grow under high-temperature and -humidity conditions and had the ability of a high lipase production. In addition, lipase derived from yeast had higher safety and was widely used in various industries such as food processing compared to bacterial lipase. The production process of lipases derived from these microbials was also more environmentally friendly, using renewable raw materials and biodegradable reagents, thereby reducing the impact on the environment.

*Y. lipolytica* is an unconventional yeast that has attracted attention due to its strong ability to decompose lipids and proteins. Its wild type could usually be isolated from environments rich in lipids or proteins, as well as from sewage- or oil-contaminated water bodies, and this species was widely distributed in extreme natural and human environments [20]. Compared to other yeasts, *Y. lipolytica* is a strict aerobic bacterium, and oxygen concentration was a key factor in its growth. The optimal growth temperature for this strain was between 25 and 30 °C, with a maximum of 34 °C. It could grow in environments with pH values ranging from 3.5 to 8.0, and a few strains could tolerate lower pH values (2.0) or higher pH values (9.7). Due to its presence in saline environments, most strains could tolerate a salt concentration of 7.5%, with some strains even reaching a tolerance of 15% [21].

In order to improve the efficiency of lipase production, *Y. lipolytica*'s technology was widely used. The fermentation process could be divided into two types: submerged fermentation (SMF) and solid-state fermentation (SSF). Submerged fermentation referred to the use of liquid culture medium for microbial fermentation, which produced approximately 90% of industrial lipase. This study also adopted a submerged fermentation method. Additionally, the process of producing *Y. lipolytica* lipase by fermentation was influenced

by various factors including carbon source, nitrogen source, carbon/nitrogen ratio, culture temperature, pH value and dissolved oxygen [22].

This article aims to address the following scientific issues. (1) In order to improve the hydrolytic enzyme activity of lipase, a series of optimizations for carbon and nitrogen sources were carried out based on the laboratory lipase strains L11-01. By continuously adjusting the culture medium formula and optimizing the culture conditions, a more efficient lipase production method was ultimately obtained. (2) In order to further improve the level of lipase fermentation, a fermentation amplification experiment was conducted: continuously cultivate L11-01 using a 5 L fermentation tank, providing stable growth conditions by controlling speed, ventilation and pH. We fully amplified and verified the regulation of shaking flask fermentation, integrated regulation methods and optimized carbon and nitrogen sources comprehensively. (3) In order to obtain a good conversion rate of liquid biofuel precursors, high-activity lipase mentioned above was used to produce free fatty acids. Reaction conditions were optimized and controlled in a self-made static reactor.

## 2. Materials and Methods

### 2.1. Materials

Malt soaking powder, yeast soaking powder, peptone, D-glucose, potassium hydroxide, phosphate, phosphoric acid, ammonia, boric acid, sodium hydroxide, polyvinyl alcohol 1750, triethylamine, phenyl isothiocyanate, sodium acetate, glacial acetic acid, citric acid monohydrate and erythritol were sourced from the National Pharmaceutical Group Chemical Reagent Co., Ltd., (Shanghai, China). Agar powder was supplied by the Aoboxing Biotechnology Co., Ltd., (Beijing, China). Calcium sulfate dihydrate, potassium sulfate, magnesium sulfate heptahydrate, manganese sulfate monohydrate, disodium ethylenediaminetetraacetic acid, zinc sulfate heptahydrate, manganese chloride tetrahydrate, cobalt chloride hexahydrate, sodium molybdate dihydrate, ferrous sulfate heptahydrate, calcium chloride dihydrate, potassium iodide, copper sulfate pentahydrate, biotin, calcium pantothenate, niacin, inositol, vitamin $B_6$, potassium dihydrogen phosphate trihydrate, histidine, aspartic acid, lysine and tyrosine were produced by the Yuanye Biotechnology Co., Ltd., (Shanghai, China). Triamine citrate, D-sorbitol, vitamin $B_1$, amino-free yeast nitrogen source, olive oil, rhodamine B, phenolphthalein and acetonitrile were purchased from the McLean Biochemical Technology Co., Ltd., (Shanghai, China). Thirty-seven mixed standards of fatty acid methyl ester used in analysis were purchased from the Sigma-Aldrich Co., Ltd., (Darmstadt, German). Fulinmen Grade 1 Soybean Oil, whose composition was mentioned in our previous research work [23], was sourced from the COFCO Group Co., Ltd., (Beijing, China). Its average fatty acid molecular weight was estimated to be 872.39 g/mol according to its fatty acid composition (Table 1) determined by the in situ methanolysis method [24]. Methanol was purchased from the Beijing Chemical Plant (Beijing, China). The above reagents were all analytical grade. Deionized water was produced in the laboratory.

**Table 1.** Chemical composition of refined soybean oil.

| Fatty Acid Molecular Formula | Types of Fatty Acids | Composition (wt%) |
| :---: | :---: | :---: |
| C16:0 | Palmitic acid | 11.56 |
| C18:0 | Stearic acid | 2.83 |
| C18:1 | Oleic acid | 20.44 |
| C18:2 | Linoleic acid | 56.15 |
| C18:3 | Linolenic acid | 8.68 |
| C20:0 | Arachidic acid | 0.34 |

### 2.2. GC Method for Determining Free Fatty Acids

A gas chromatograph (Thermo Fisher Scientific, Waltham, MA, USA, Trace 1300, Agilent Technologies, Santa Clara, CA, USA) with a DB-1HT capillary column (30 m × 0.25 mm × 0.10 mm, Agilent Technologies, CA, USA) was used to determine

the content of each component in the reaction system. When measuring the sample, the heating program for capillary columns was: Column temperature rises from 100 °C to 180 °C at 15 °C/min, held for 5 min, and then rises from 180 °C to 350 °C at 20 °C/min, held for 15 min. The temperatures of the injector and detector were set at 350 °C. The peaks of free fatty acids were determined by their retention time, using two standards: pentadecanoic and heptadecanoic acid isobutyl esters, purchased from Sigma, were used as the internal standard. The FFAs yield was defined as generated FFAs amount during the reaction divided by the initial soybean oil amount ($g/g$).

### 2.3. Shake Flask Fermentation

Add 50 mL of fermentation medium (shown in SI) to a 200 mL shaker, seal with 16 layers of gauze, and reseal with newspaper. Sterilize under high pressure at 116 °C for 25 min, cool, and then add 1 mL of the secondary seed solution (shown in SI). Cultivate in a shaker at 30 °C and 200 rpm. Add 0.8% ($v/v$) trace element solution and vitamin solution based on the volume of the fermentation medium at 0 and 24 h, and add 8% ($v/v$) soybean oil based on the volume of the fermentation medium as a carbon source every 24 h.

#### 2.3.1. The Impact of the Citric Acid Addition in a Shaking Flask

The fermentation time for the subsequent shaking flask experiment was set to 120 h (shown in Figure S2). Through the analysis of the metabolic pathway of *Y. lipolytica*, citric acid, an intermediate in the tricarboxylic acid cycle, was selected as an additive for ATP supply, providing more energy for strains' growth and enzyme production. Citric acid, as the central metabolite in the metabolic pathway of *Y. lipolytica*, was closely related to lipid metabolism and synthesis pathways. The impact of citric acid on the initial growth of the strains and hydrolase activity is shown in Section 3.1.1.

#### 2.3.2. The Impact of the Types and Amounts of Amino Acids in a Shaking Flask

The nitrogen source has a significant impact on lipase activity. In order to accurately regulate the nitrogen source during the fermentation process, the exploration of composite amino acids was conducted. The optimization of a composite amino acid shaking flask aimed to maximize the activity of hydrolytic enzymes while maintaining similar growth conditions. This experiment explored the types and amounts of amino acids added during the fermentation process in Section 3.1.2 and conducted a scaled-up validation test on a 5 L fermentation tank in Section 3.2.3. Quantitative analysis of 18 different amino acids was shown in Figure S1, in which the mobile phase gradient was shown in Table S1.

### 2.4. Fermentation Process of 5 L Fermentation Tank

The maximum activity of lipase hydrolase produced by shaking flask fermentation reached 2200 U/mL. In order to further improve the level of lipase fermentation, a fermentation amplification experiment was conducted, by continuously cultivating *Y. lipolytica* L11-01 using a 5 L fermentation tank, providing stable growth conditions by controlling speed, ventilation and pH. We amplified and verified the regulation of shaking flask fermentation in Section 3.1, integrated regulation methods and optimized carbon and nitrogen sources comprehensively.

Prepare 2.5 L of fermentation medium, sterilize and use ammonia or 85% phosphoric acid to control the pH to 4.8; subsequent adjustments are made through pH electrode detection. Set the pH value of the fermentation tank system to 4.8. Ammonia is automatically added to maintain a pH of 4.8 when the pH decreased during the fermentation process. Two bottles (200 mL) of secondary seed solution are added to the fermentation tank culture medium with an inoculation amount of 8%. Add 0.8% ($v/v$) trace element solution and vitamin solution based on the volume of the culture medium at 0 and 24 h, and manually supplement 8% ($v/v$) soybean oil based on the volume of the culture medium every 24 h.

### 2.4.1. The Impact of the Citric Acid Addition in a 5 L Fermentation Tank

Based on the results of the shaking flask's citric acid optimization experiment, the amplification validation was carried out in a 5 L fermentation tank. In order to prevent the onetime addition of a large amount of citric acid from affecting strains' growth, it was decided to add it in batches, and the pH in the fermentation tank was maintained at 4.8 using ammonia throughout the fermentation process. This not only supplemented carbon sources but also nitrogen sources. The experimental results can be found in Section 3.2.1.

### 2.4.2. The Impact of the Carbon Source Automatic Replenishment Scheme in a 5 L Fermentation Tank

Soybean oil not only serves as a carbon source in fermentation, but also induces the expression of lipase. The initial tank loading process involves manually adding soybean oil every day. In order to further regulate the carbon source feeding and design an automatic feeding scheme, the consumption and its rate of soybean oil in a 5 L fermentation tank were measured. The experimental results can be found in Section 3.2.2.

### 2.4.3. The Impact of the Compound Amino Acids in a 5 L Fermentation Tank

Based on the above results of the composite amino acid optimization experiment in the shaking flask, the amplification validation was carried out in a 5 L fermentation tank. A total of 4 schemes of 5 L fermentation tank fermentation were carried out. The first option was not to add amino carboxylic acids, the second option was to add 0.1 mol/L of histidine at the beginning of fermentation, the third option was to add 0.1 mol/L of aspartic acid at the beginning of fermentation and the fourth option was to add composite amino acids: 0.1 mol/L of histidine, 0.03 mol/L of aspartic acid and 0.03 mol/L of lysine at the beginning of fermentation. We conducted a 312 h fermentation experiment and monitored the biomass $OD_{600nm}$ and hydrolytic enzyme activity during the fermentation process every 24 h. The experimental results are shown in Section 3.2.3.

### 2.4.4. Multibatch Stable Fermentation in a 5 L Fermentation Tank

After adding citric acid, carbon source automatic feeding and composite amino acid three-step optimization, the hydrolytic enzyme activity in a 5 L fermentation tank experienced a progressive increase process, from 9750 U/mL to 13,625 U/mL, then to 20,220 U/mL and finally to 26,800 U/mL. In order to verify the authenticity of the experimental results, the above three optimization steps were combined for multiple fermentation, and the biomass and hydrolytic enzyme activity of the fermentation broth were measured at 300 h. The experimental results are shown in Section 3.2.4.

### 2.5. Lipase Hydrolysis Technology of Soybean Oil

The process experiment of lipase hydrolysis of soybean oil is carried out in a self-designed static reactor in the laboratory. Firstly, the water bath temperature is raised to 30–55 °C to heat the reaction device. Then, 1200 g of soybean oil are added to the oil tank, and a peristaltic pump is started to quickly preheat the raw materials to the corresponding temperature. Finally, 1–5% (*w/w*) lipase enzyme and 5–35% (*w/w*) deionized water based on the soybean oil weight are added to the tank to start the timed reaction.

On the basis of optimizing the culture conditions of *Y. lipolytica* and regulating the efficient expression of lipase in a 5 L fermentation tank, a lipase fermentation broth with a laboratory hydrolytic enzyme activity of 25,000 U/mL was applied to a self-made static reactor in the laboratory to hydrolyze soybean oil into free fatty acids. In order to achieve a high yield of free fatty acids, various reaction parameters (material flow rate, water content, reaction temperature, lipase fermentation broth content and reuse of lipase fermentation broth) in the whole process were optimized.

2.5.1. The Impact of the Material Flow Rate on the Production of Free Fatty Acids

The self-made static reactor in our laboratory was different from the conventional stirred reactor. It strengthened the mass transfer and mixing rate between soybean oil, lipase fermentation broth and deionized water through the self-designed static mixer, and the mixing uniformity in the static mixer was closely related to the material flow rate [23]. When the material flow rate was low, the oil and water phases were not only easy to separate due to poor mixing, but also reduced the heat transfer to the material in a timely and rapid manner. This affected the reaction rate and the final product yield. Therefore, material flow rate was one of the most important process parameters in the production of free fatty acids. The experimental results are shown in Section 3.3.1.

2.5.2. The Impact of the Reaction Temperature on the Production of Free Fatty Acids

Another important parameter in the production process of free fatty acids is the reaction temperature. Lower temperatures not only affect the viscosity and physical state of the material, but also lower the molecular motion, resulting in a lower reaction rate overall. Although excessive temperature could accelerate molecular motion, it had a more significant deactivation effect on lipase and also increased energy consumption [25]. Therefore, selecting an appropriate reaction temperature is crucial for the lipase hydrolysis of soybean oil to produce free fatty acids. The experimental results are shown in Section 3.3.2.

2.5.3. The Impact of the Water Content on the Production of Free Fatty Acids

In enzymatic hydrolysis reactions, water is not only the reactant, but also directly affects the equilibrium rate and reaction degree of the chemical reaction. At the same time, water together with oil provides an oil–water interface to activate the lipase. The hydrophilic group of the lipase faces the water side, and the hydrophobic group faces the oil side. In addition, water was also the solvent for lipase, and excessive water would dilute the concentration of lipase [26]. Therefore, further optimization of water content is needed. The experimental results are shown in Section 3.3.3.

2.5.4. The Impact of the Lipase Fermentation Broth Dosage on the Production of Free Fatty Acids

The production cost of free fatty acids comes partly from the cost of raw materials and partly from the cost of catalyst lipase. The cost of lipase is not only related to the adjustment of fermentation medium formula and the optimization of 5 L fermentation tank regulation, but also to the dosage and repeated use of lipase [27,28]. It is necessary to investigate the amount of lipase used. The experimental results are shown in Section 3.3.4.

2.5.5. The Impact of the Reuse of Lipase Fermentation Broth on the Production of Free Fatty Acids

As a part of the production cost, catalysts needed to be aligned and reused after use. Meanwhile, considering that water had no adverse effect on lipase fermentation broth, water and lipase fermentation broth were separated and recycled together after centrifugation. The experimental results are shown in Section 3.3.5.

## 3. Results and Discussion

*3.1. Adjusting the Medium Formula and Optimizing Cultivation Conditions*

3.1.1. Optimization of Citric Acid Addition in a Shaking Flask

In order to reduce the impact of citric acid on the initial growth of the strains, 3, 6, 9, 12, 15 and 18 g/L of citric acid were added during 24 h of fermentation, and the biomass $OD_{600nm}$ and hydrolytic enzyme activity during the fermentation process were detected every 24 h. The experimental results are shown in Figure 1.

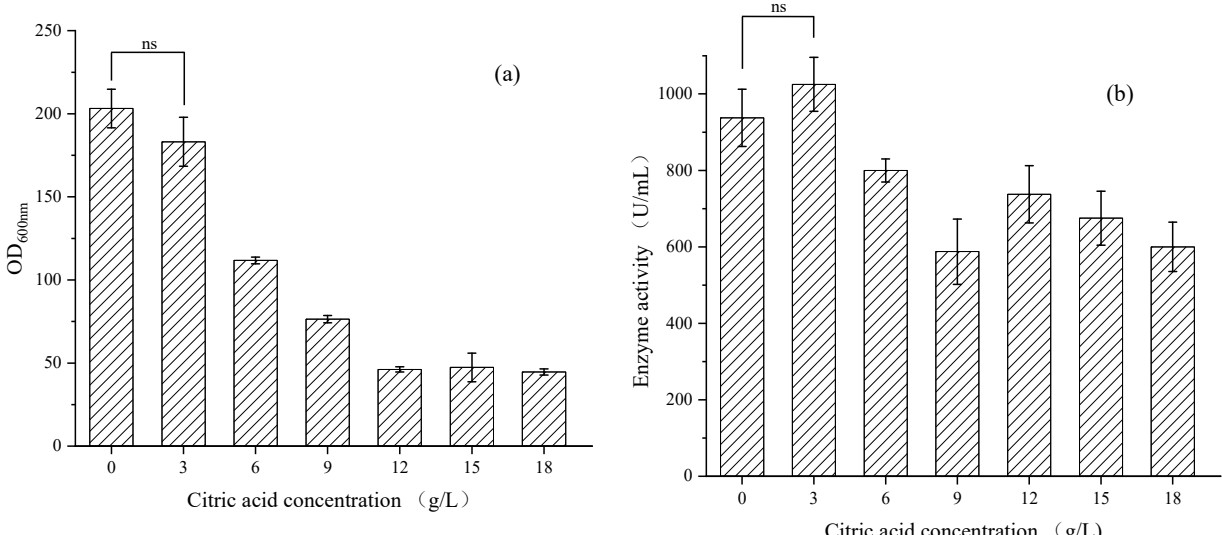

**Figure 1.** (**a**) Biomass versus citric acid concentration gradient during 120 h of shaking flask fermentation; (**b**) Hydrolase activity versus citric acid concentration gradient during 120 h of shaking flask fermentation. The data represent the mean ± standard deviation (*n* = 3). Statistical analysis was carried out by using the Student's *t* test (one-tailed, ns, not significant).

Adding low-concentration citric acid promoted the growth of the strains and provided additional NADPH for the lipid metabolism pathway, thereby promoting the expression level of lipase and the hydrolytic activity. When adding approximately 3 g/L of citric acid for 120 h, the hydrolytic enzyme activity was 1025 U/mL, which was 109% of the control group. This was consistent with the results of the significance test shown in Figure 1, which shows that 3 g/L of citric acid had no significant effect on the growth of the strains and enzyme activity. However, the inhibitory effect continued to increase as the concentration of citric acid increased, and it was particularly evident within 24 h after addition. The activity of the hydrolytic enzyme hardly increased. When adding approximately 18 g/L of citric acid for 120 h, the activity of the hydrolytic enzyme was 600 U/mL, only 64% of the control group. The reason was that high concentrations of citric acid further inhibited the growth of the strains, leading to a decrease in the overall expression of lipase.

Adding 3 g/L of citric acid might inhibit the growth of the strains to a certain extent, but it promoted the expression of lipase and the activity of hydrolytic enzymes. The next step was to verify the fermentation amplification experiment in Section 3.2.1.

3.1.2. Amino Acid Screening and Optimization of Composite Amino Acids
Pre-Experiment for Screening 20 Amino Acids

Firstly, an amino acid screening pre-experiment was conducted, and a total of 20 amino acids were selected: glycine, alanine, valine, leucine, isoleucine, methionine, proline, tryptophan, serine, tyrosine, cysteine, phenylalanine, asparagine, glutamine, threonine, aspartic acid, glutamic acid, lysine, arginine and histidine. We added 0.09 mol/L of different amino acids to the initial fermentation medium and detected the biomass $OD_{600nm}$ and hydrolytic enzyme activity every 24 h during the fermentation process. The experimental results are shown in Figure 2.

The amino acids that had a promoting effect on the biomass of the strains were: histidine > glycine > alanine > control group.

The amino acids that had a promoting effect on the hydrolytic enzyme activity of the strains were: tyrosine > lysine > aspartic acid = histidine > glutamine > glutamic acid > glycine > control group.

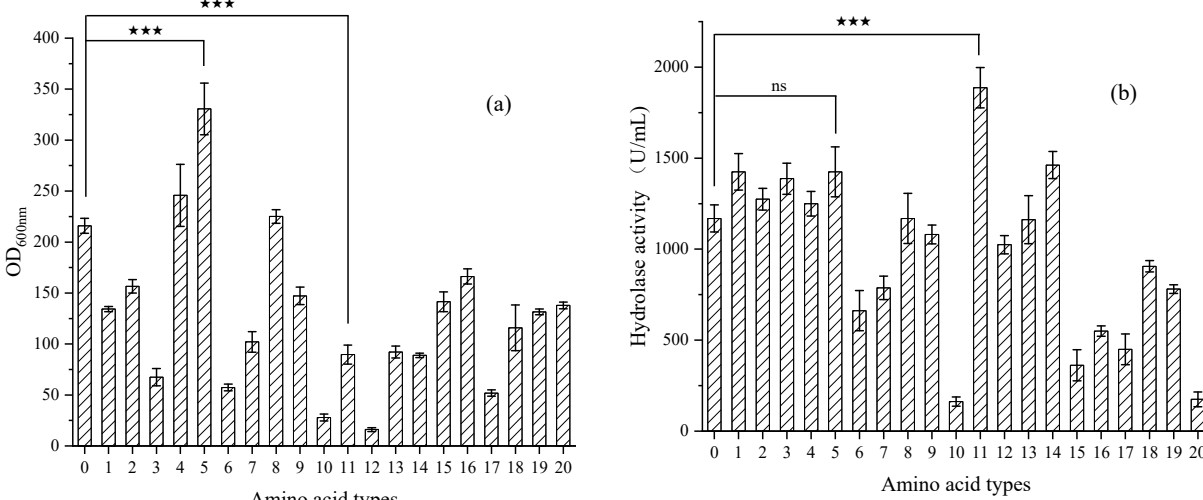

**Figure 2.** (**a**) Biomass versus different types of amino acids during 120 h of shaking flask fermentation; (**b**) Hydrolase activity versus different types of amino acids during 120 h of shaking flask fermentation. The data represent the mean $\pm$ standard deviation ($n$ = 3). Statistical analysis was carried out by using the Student's $t$ test (one-tailed, ns, not significant; *** $p < 0.001$).

When adding approximately 0.09 mol/L of tyrosine, the hydrolytic enzyme activity at 120 h was approximately 1887 U/mL, which was 162% of the control group and had a strong promoting effect on lipase expression. However, the biomass $OD_{600nm}$ at 120 h was approximately 89.69, which was only 41% of the control group. Through experiments, it was found that most amino acids could only reflect one aspect of improving strains' growth or increasing lipase expression and might even have inhibitory effects on the other. Therefore, it was considered to combine several amino acids to explore their combined effects.

After screening, it was found that histidine had a good promoting effect on both aspects, the biomass of the strains in the histidine group was 153% of that in the control group, and the hydrolytic enzyme activity in the histidine group was 122% of that in the control group. When adding approximately 0.09 mol/L of histidine, the 120 h biomass $OD_{600nm}$ was approximately 330.60, and the activity of the hydrolytic enzyme was approximately 1425 U/mL.

Optimization Experiment of Four Amino Acid Concentration Gradients

Based on the above preliminary experimental results, the concentration of single amino acids was optimized: selected histidine with good overall effect, aspartic acid, lysine and tyrosine with significant promoting effect on the activity of hydrolytic enzymes to optimize the concentration of single amino acids. Four amino acids with concentrations of 0.05, 0.1, 0.15, 0.2 and 0.25 mol/L, respectively, were added and detected the biomass $OD_{600nm}$ and hydrolytic enzyme activity during the fermentation process every 24 h. The experimental results are shown in Figure 3.

In addition to adding histidine, the addition of aspartic acid, lysine and tyrosine could all inhibit the growth of the strains to varying degrees, and most amino acid concentrations would delay the growth period of the strains. Although adding histidine could promote the growth of the strains, the higher the concentration, the less significant the effect. The highest biomass group for each amino acid after 120 h of shaking flask fermentation was as follows: when 0.05 mol/L histidine was added, the biomass $OD_{600nm}$ at 120 h was approximately 329.18, which was 166% of the control group; when 0.15 mol/L aspartic acid was added, the biomass $OD_{600nm}$ for 120 h was approximately 165.5; when adding 0.20 mol/L lysine for 120 h, the biomass $OD_{600nm}$ was approximately 189.60; when adding 0.25 mol/L tyrosine for 120 h, the biomass $OD_{600nm}$ was approximately 182.80.

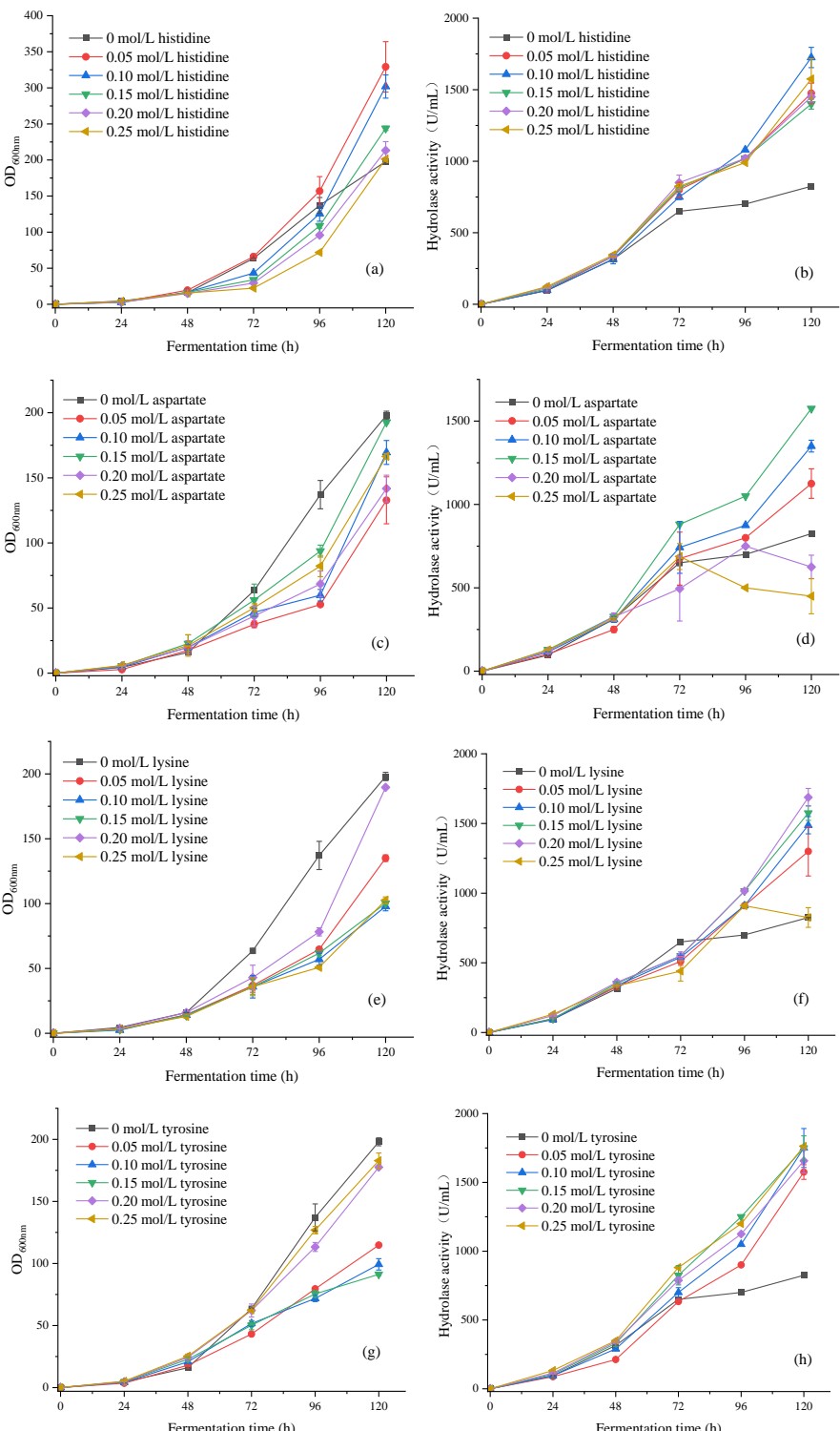

**Figure 3.** (**a**) The biomass of histidine concentration gradient during 120 h of shaking flask fermentation; (**b**) The hydrolytic enzyme activity of histidine concentration gradient during 120 h of shaking flask fermentation; (**c**) The biomass of aspartic acid concentration gradient during 120 h of shaking flask fermentation; (**d**) The hydrolytic enzyme activity of aspartic acid concentration gradient during 120 h of shaking flask fermentation; (**e**) The biomass of lysine concentration gradient during 120 h of shaking flask fermentation; (**f**) The hydrolytic enzyme activity of lysine concentration gradient during 120 h of shaking flask fermentation; (**g**) The biomass of tyrosine concentration gradient during 120 h of shaking flask fermentation; (**h**) The hydrolytic enzyme activity of tyrosine concentration gradient during 120 h of shaking flask fermentation.

Adding 0.2, 0.25 mol/L aspartic acid or 0.25 mol/L lysine could inhibit the activity of hydrolase and even led to a decrease in enzyme activity after 96 h of fermentation. In addition, different concentrations of four amino acids could promote the expression of lipase. The highest group of hydrolytic enzyme activity for each amino acid after 120 h of shaking flask fermentation was as follows: when 0.10 mol/L histidine was added, the hydrolytic enzyme activity for 120 h was approximately 1725 U/mL; the hydrolytic enzyme activity was approximately 1575 U/mL after 120 h of adding 0.15 mol/L aspartic acid; when adding 0.20 mol/L lysine for 120 h, the hydrolytic enzyme activity was approximately 1687 U/mL; when adding 0.25 mol/L tyrosine, the hydrolytic enzyme activity at 120 h was approximately 1762 U/mL, which was 213% of the control group.

In order to balance strains' growth and lipase expression, histidine was selected as the preferred amino acid for optimization, and aspartic acid, lysine and tyrosine were sequentially added for combined amino acid optimization. Considering the overlapping metabolic pathways of amino acids, response surface optimization was not used in the subsequent experimental plan, but orthogonal optimization was used, and the subsequent amino acid addition was appropriately reduced.

Composite Optimization of Two Amino Acids

Considering that different amino acid metabolic pathways might be the same, concentration gradient optimization was chosen. Based on the above experimental results, the concentration optimization of two types of amino acids was carried out: 15 combinations were selected with histidine concentrations of 0.05, 0.1 and 0.15 mol/L, and aspartic acid concentrations of 0.03, 0.06, 0.09, 0.12 and 0.15 mol/L. A 120 h fermentation experiment was conducted, and the biomass $OD_{600nm}$ and hydrolytic enzyme activity during the fermentation process were detected every 24 h. The experimental results are shown in Table 2.

**Table 2.** Optimization of shaking flask fermentation biomass and hydrolytic enzyme activity at 120 h with different concentrations of histidine and aspartic acid.

| Serial Number | A | B | $OD_{600nm}$ | Hydrolase Activity (U/mL) |
| --- | --- | --- | --- | --- |
| | Histidine (mol/L) | Aspartate (mol/L) | | |
| 1 | 0.05 | 0 | 329.18 ± 34.90 | 1475.0 ± 70.71 |
| 2 | 0.05 | 0.03 | 113.13 ± 4.42 | 1520.0 ± 167.94 |
| 3 | 0.05 | 0.06 | 166.88 ± 5.83 | 1957.5 ± 20.89 |
| 4 | 0.05 | 0.09 | 179.40 ± 1.70 | 1867.5 ± 68.10 |
| 5 | 0.05 | 0.12 | 116.25 ± 4.95 | 1710.0 ± 29.64 |
| 6 | 0.05 | 0.15 | 104.00 ± 3.54 | 1665.0 ± 25.68 |
| 7 | 0.1 | 0 | 301.88 ± 16.09 | 1725.0 ± 58.96 |
| 8 | 0.1 | 0.03 | 200.00 ± 3.98 | 1912.5 ± 127.28 |
| 9 | 0.1 | 0.06 | 165.63 ± 13.26 | 1822.5 ± 20.64 |
| 10 | 0.1 | 0.09 | 133.13 ± 8.31 | 1710.0 ± 95.46 |
| 11 | 0.1 | 0.12 | 96.50 ± 1.06 | 1417.5 ± 63.64 |
| 12 | 0.1 | 0.15 | 108.00 ± 5.30 | 1507.5 ± 25.98 |
| 13 | 0.15 | 0 | 243.95 ± 2.97 | 1400.0 ± 35.36 |
| 14 | 0.15 | 0.03 | 173.00 ± 3.18 | 1665.0 ± 30.58 |
| 15 | 0.15 | 0.06 | 140.00 ± 1.58 | 1665.0 ± 98.59 |
| 16 | 0.15 | 0.09 | 117.13 ± 7.25 | 1620.0 ± 31.82 |
| 17 | 0.15 | 0.12 | 128.88 ± 4.54 | 1462.5 ± 65.49 |
| 18 | 0.15 | 0.15 | 118.88 ± 1.59 | 1507.5 ± 60.98 |

When histidine and aspartic acid were added simultaneously, it greatly inhibited and affected the growth of strains. However, it was found that adding 0.1 mol/L histidine and 0.03 mol/L aspartic acid for fermentation for 120 h could increase the biomass $OD_{600nm}$ to 200, slightly higher than the group without adding amino acids after screening with different levels of addition.

On the contrary, the expression level of lipase also increased to varying degrees when histidine and aspartic acid were added simultaneously. This indicated that, although adding composite amino acids reduced the number of bacterial cells, it greatly improved the enzyme production efficiency per unit cell, which was basically in line with the optimization purpose of this experiment.

Four groups of combinations with higher hydrolytic enzyme activity were selected for subsequent experiments: (1) adding 0.05 mol/L histidine and 0.06 mol/L aspartic acid, hydrolytic enzyme activity was 1975 U/mL, which was 132% of the control group after 120 h of shaking flask fermentation; (2) adding 0.05 mol/L histidine and 0.09 mol/L aspartic acid, hydrolytic enzyme activity was 1867 U/mL, which was 127% of the control group after 120 h of shaking flask fermentation; (3) adding 0.1 mol/L histidine and 0.03 mol/L aspartic acid, hydrolytic enzyme activity was 1912 U/mL, which was 111% of the control group after 120 h of shaking flask fermentation; (4) adding 0.1 mol/L histidine and 0.06 mol/L aspartic acid, hydrolytic enzyme activity was 1822 U/mL, which was 106% of the control group after 120 h of shaking flask fermentation.

Optimization of Three Composite Amino Acids

Based on the above experimental results, three amino acid co-addition concentrations were optimized: 0.03, 0.06, 0.09 and 0.12 mol/L lysine were added to two composite amino acid culture media with a total of 16 combinations. A 120 h fermentation experiment was conducted, and the biomass $OD_{600nm}$ and hydrolytic enzyme activity were detected every 24 h during the fermentation process. The experimental results are shown in Table 3.

**Table 3.** Biomass and hydrolase activity of different concentrations of histidine, aspartic acid and lysine during 120 h combinatorial optimization shake flask fermentation.

| Serial Number | A | | B | $OD_{600nm}$ | Hydrolase Activity (U/mL) |
|---|---|---|---|---|---|
| | Histidine (mol/L) | Aspartate (mol/L) | Lysine (mol/L) | | |
| 1 | 0.05 | 0.06 | 0 | 166.88 ± 5.83 | 1957.50 ± 20.89 |
| 2 | 0.05 | 0.06 | 0.03 | 86.25 ± 0.67 | 1656.25 ± 128.21 |
| 3 | 0.05 | 0.06 | 0.06 | 86.50 ± 0.34 | 1750.00 ± 93.60 |
| 4 | 0.05 | 0.06 | 0.09 | 107.88 ± 0.84 | 2242.50 ± 140.51 |
| 5 | 0.05 | 0.06 | 0.12 | 95.25 ± 2.79 | 2031.25 ± 126.88 |
| 6 | 0.05 | 0.09 | 0 | 179.40 ± 1.70 | 1867.50 ± 68.10 |
| 7 | 0.05 | 0.09 | 0.03 | 139.50 ± 2.77 | 1906.25 ± 43.22 |
| 8 | 0.05 | 0.09 | 0.06 | 31.25 ± 0.69 | 1125.00 ± 175.01 |
| 9 | 0.05 | 0.09 | 0.09 | 37.50 ± 1.37 | 1312.50 ± 84.76 |
| 10 | 0.05 | 0.09 | 0.12 | 77.38 ± 0.86 | 1375.00 ± 86.71 |
| 11 | 0.1 | 0.03 | 0 | 200.00 ± 3.98 | 1912.50 ± 127.28 |
| 12 | 0.1 | 0.03 | 0.03 | 188.75 ± 0.35 | 2117.50 ± 112.94 |
| 13 | 0.1 | 0.03 | 0.06 | 151.50 ± 1.03 | 2031.25 ± 42.21 |
| 14 | 0.1 | 0.03 | 0.09 | 61.38 ± 1.26 | 1843.75 ± 44.81 |
| 15 | 0.1 | 0.03 | 0.12 | 113.13 ± 1.66 | 2093.75 ± 45.65 |
| 16 | 0.1 | 0.06 | 0 | 165.63 ± 13.26 | 1822.5 ± 20.64 |
| 17 | 0.1 | 0.06 | 0.03 | 143.63 ± 0.93 | 2127.50 ± 79.77 |
| 18 | 0.1 | 0.06 | 0.06 | 124.50 ± 2.87 | 1906.25 ± 46.09 |
| 19 | 0.1 | 0.06 | 0.09 | 112.13 ± 0.54 | 1906.25 ± 45.61 |
| 20 | 0.1 | 0.06 | 0.12 | 111.13 ± 5.96 | 1875.00 ± 18.07 |

When histidine, aspartic acid and lysine were added simultaneously, it would also inhibit and affect the growth of strains. However, after screening with different levels of addition, it was found that adding 0.1 mol/L histidine, 0.03 mol/L aspartic acid and 0.03 mol/L lysine during fermentation for 120 h resulted in a biomass $OD_{600nm}$ of approximately 188, which was 95% of the group without the addition of amino acids.

The addition of different concentrations of lysine had little effect on the hydrolytic enzyme activity after 120 h of shaking flask fermentation, but a significant increase in

enzyme activity could be observed after 96 h. It was expected that extending the cultivation time would result in a significant increase in enzyme activity, which needed to be verified in a subsequent 5 L fermentation tank.

Three groups of combinations with higher hydrolytic enzyme activity were selected for subsequent experiments: (1) adding 0.05 mol/L histidine, 0.06 mol/L aspartic acid and 0.09 mol/L lysine, a hydrolytic enzyme activity was 2242 U/mL, which was 114% of the control group; (2) adding 0.1 mol/L histidine, 0.03 mol/L aspartic acid and 0.03 mol/L lysine, a hydrolytic enzyme activity was 2213 U/mL, which was 115% of the control group; (3) adding 0.1 mol/L histidine, 0.06 mol/L aspartic acid and 0.03 mol/L lysine, a hydrolytic enzyme activity was 2127 U/mL, which was 117% of the control group.

Optimization of Four Compound Amino Acids

Based on the above experimental results, four types of amino acids were optimized for co-addition concentration: 0.03, 0.06, 0.09 and 0.12 mol/L tyrosine were added to three composite amino acid culture media with a total of 12 combinations. A 120 h fermentation experiment was conducted, and the biomass $OD_{600nm}$ and hydrolytic enzyme activity were monitored every 24 h during the fermentation process. The experimental results are shown in Table 4.

**Table 4.** Biomass and hydrolase activity of different concentrations of histidine, aspartic acid, lysine and tyrosine during 120 h of combinatorial optimization shake flask fermentation.

| Serial Number | A | | | B | $OD_{600nm}$ | Hydrolase Activity (U/mL) |
| --- | --- | --- | --- | --- | --- | --- |
| | Histidine (mol/L) | Aspartate (mol/L) | Lysine (mol/L) | Tyrosine (mol/L) | | |
| 1 | 0.05 | 0.06 | 0.09 | 0 | 107.88 ± 0.84 | 2242.5 ± 140.51 |
| 2 | 0.05 | 0.06 | 0.09 | 0.03 | 86.25 ± 0.69 | 1957.5 ± 30.36 |
| 3 | 0.05 | 0.06 | 0.09 | 0.06 | 86.50 ± 0.34 | 1755.0 ± 44.14 |
| 4 | 0.05 | 0.06 | 0.09 | 0.09 | 107.88 ± 0.85 | 1777.5 ± 90.59 |
| 5 | 0.05 | 0.06 | 0.09 | 0.12 | 95.25 ± 2.70 | 1800.0 ± 58.90 |
| 6 | 0.1 | 0.03 | 0.03 | 0 | 188.75 ± 0.35 | 2117.5 ± 112.94 |
| 7 | 0.1 | 0.03 | 0.03 | 0.03 | 139.50 ± 2.69 | 1890.0 ± 62.24 |
| 8 | 0.1 | 0.03 | 0.03 | 0.06 | 31.25 ± 0.68 | 1552.5 ± 155.12 |
| 9 | 0.1 | 0.03 | 0.03 | 0.09 | 37.50 ± 1.37 | 1687.5 ± 30.87 |
| 10 | 0.1 | 0.03 | 0.03 | 0.12 | 77.38 ± 0.85 | 1710.0 ± 25.85 |
| 11 | 0.1 | 0.06 | 0.03 | 0 | 151.50 ± 1.03 | 2031.25 ± 42.21 |
| 12 | 0.1 | 0.06 | 0.03 | 0.03 | 188.75 ± 0.34 | 1800.0 ± 124.86 |
| 13 | 0.1 | 0.06 | 0.03 | 0.06 | 151.50 ± 1.03 | 1732.5 ± 30.77 |
| 14 | 0.1 | 0.06 | 0.03 | 0.09 | 61.38 ± 1.29 | 1665.0 ± 64.59 |
| 15 | 0.1 | 0.06 | 0.03 | 0.12 | 113.13 ± 1.62 | 1642.5 ± 99.85 |

After adding four types of amino acids, the growth of most strains in the experimental group was relatively severe, which could lead to unpredictable effects in subsequent 5 L fermentation tank scale-up experiments. Therefore, adding tyrosine to the composite amino acids was not considered.

In addition, adding tyrosine could not further increase the expression level of lipase, but would have a counterproductive effect, leading to a decrease in the activity of hydrolases. Similarly, adding tyrosine to composite amino acids was not considered.

In summary, after conducting a series of composite amino acid optimization experiments mentioned above, the best-performing combination was: 0.1 mol/L histidine, 0.03 mol/L aspartic acid and 0.03 mol/L lysine. Its biomass $OD_{600nm}$ was approximately 188, which was 95% of the group without amino acids added, and its hydrolytic enzyme activity was 2213 U/mL, which was 272% of the group without amino acids added.

### 3.2. Process Optimization of Y. lipolytica in a 5 L Fermentation Tank

3.2.1. Optimization of Citric Acid Addition in a 5 L Fermentation Tank

The concentration of citric acid added each time was 5 g/L, starting from 24 h of fermentation. Subsequently, it was divided into feeding every 48 h (six times in total) and feeding every 96 h (three times in total). The experimental results are shown in Figure 4.

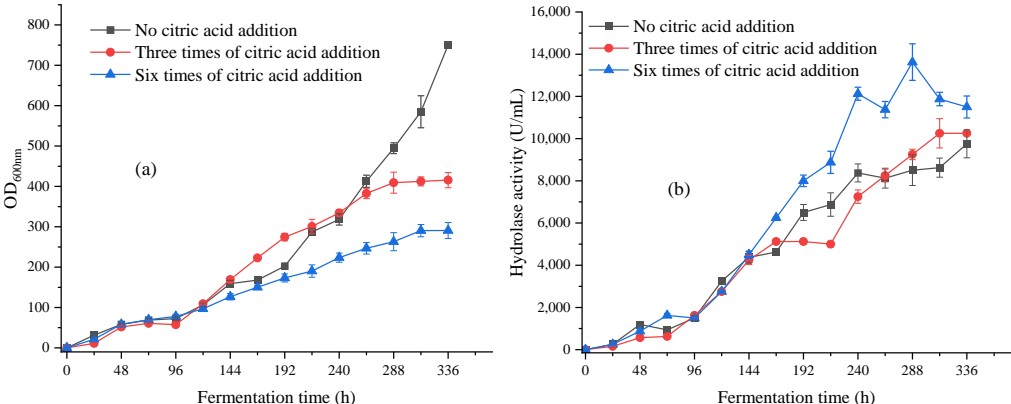

**Figure 4.** (**a**) The biomass $OD_{600nm}$ with different citric acid addition times in 336 h fermentation; (**b**) The hydrolytic enzyme activity with different citric acid addition times in 336 h fermentation.

As shown in Figure 4a, the biomass $OD_{600nm}$ of the citric acid group added three times was 415.5, which was 55% of the citric acid-free group after 336 h of fermentation; the biomass $OD_{600nm}$ of the citric acid group added six times was 290.6, which was 38% of the citric acid-free group. Compared with the control group, both groups showed a decrease in biomass, and the growth of the strains entered a stable phase after 312 h, while the control group was still in the logarithmic phase. This further confirmed the results of shaking flask fermentation and proved that citric acid had an inhibitory effect on the growth of the strains.

In Figure 4b, the group with six additions of citric acid reached the highest enzyme activity of 13,625 U/mL at 288 h, which was 140% of the group without citric acid. The enzyme activity of the citric acid-free group and the citric acid-added group after 336 h of fermentation was similar, both of which were around 10,000 U/mL. This proved that the addition of citric acid had a promoting effect on the expression of lipase, and increasing the feeding frequency of citric acid could further enhance the activity of hydrolytic enzymes.

3.2.2. Design of Carbon Source Automatic Replenishment Scheme in a 5 L Fermentation Tank

As shown in Figure 5, at 0–96 h, the consumption of soybean oil was approximately 200 g per day, with a consumption rate of approximately 8.3 g/h; the consumption of soybean oil showed a decreasing trend from 96 to 240 h; the consumption of soybean oil from 240 to 312 h was approximately 100 g per day, with a consumption rate of approximately 4.2 g/h. As the fermentation time increased, the consumption of soybean oil decreased, indicating that soybean oil was mainly used as a carbon source for the growth of the strains in the early stage of fermentation. After 240 h, the strains entered a stable period of growth, and soybean oil was mainly used as a compound to induce the strains to produce lipase.

It was hence decided to change the manual feeding mode to automatic feeding mode based on the consumption rate of soybean oil, and taking samples every 24 h for centrifugal testing of soybean oil consumption. Automatic feeding scheme: 0–96 h feeding rate 8.3 g/h; 96–240 h feeding rate 6.25 g/h; 240–312 h feeding rate 4.2 g/h. The experimental results are shown in Figure 6.

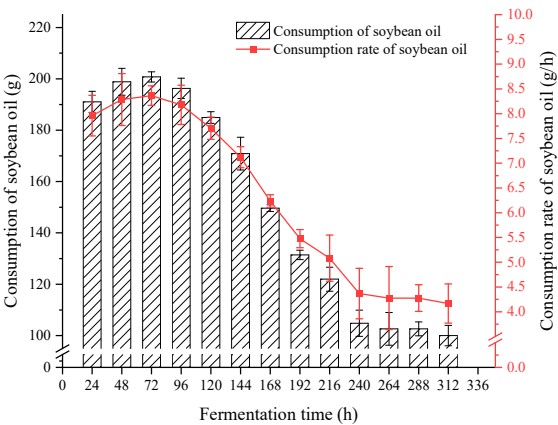

**Figure 5.** Consumption and consumption rate of manually adding soybean oil.

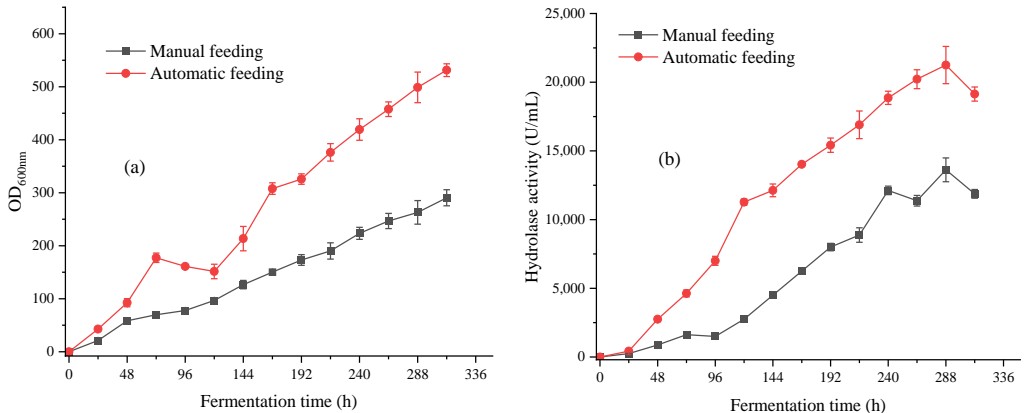

**Figure 6.** (**a**) The biomass $OD_{600nm}$ of manual and automatic feeding in 312 h fermentation; (**b**) The hydrolase activity of manual and automatic feeding in 312 h fermentation.

After adjusting the feeding method, the biomass $OD_{600nm}$ reached 530, which was 182% of the manual feeding, and the hydrolytic enzyme activity was correspondingly increased to 20,220 U/mL. Automatic feeding could improve the quality of carbon source supply, improve the growth status of strains and further increase the expression level of lipase.

### 3.2.3. Optimization of Compound Amino Acids in a 5 L Fermentation Tank

As shown in Figure 7a, the addition of 0.1 mol/L histidine significantly increased the biomass, with a maximum biomass $OD_{600nm}$ of 850 during the 312 h fermentation process, which was 65% higher than the control group. Compared to the shaking flask fermentation, adding histidine in the amplification fermentation promoted the growth of the strains to a greater extent, and there was no trend towards entering a stable growth period. However, excessive biomass could cause the fermentation broth to become increasingly viscous and make it difficult to proceed with the next step of treatment after fermentation. The addition of 0.1 mol/L aspartic acid had the opposite effect, reaching a stable period at 288 h, and the biomass was only half that of the control group. After adding composite amino acids, the growth curve of the strains was similar to that of the control group, and the composite amino acids did not have a significant impact on the growth of the strains. Except for the 0.1 mol/L histidine group, all other groups stopped growing for a period of time between 48 and 120 h. This phenomenon also occurred in the automatic feeding group in Section 3.2.2, accompanied by a significant increase in lipase hydrolase activity at the same time. This phenomenon was due to the fact that there were fewer lipases synthesized in the early stage of strains' growth, which could not hydrolyze a large amount of soybean oil, resulting in less available free fatty acids. A large amount of free fatty acids was used

to promote lipase synthesis, resulting in an insufficient carbon source for strains' growth. After 120 h, a large amount of lipase was synthesized, and strains' growth was restored. Overall, the results of the fermentation tank amplification and shaking flask experiments showed the same trend, but the extent of inhibition or promotion varied.

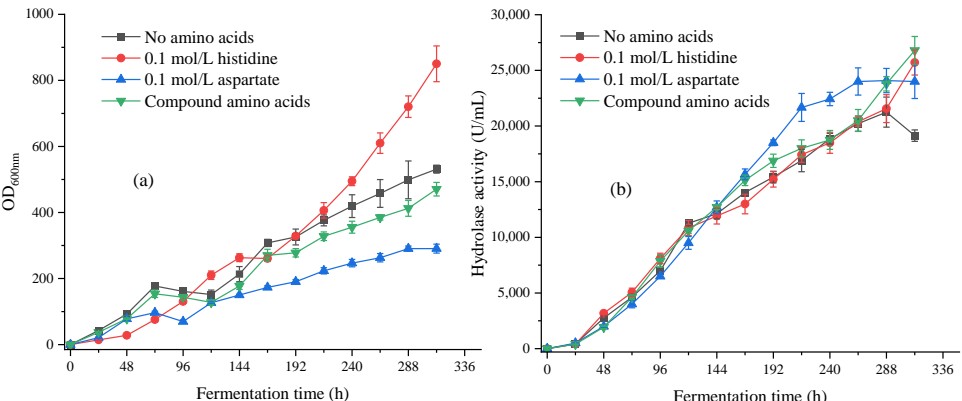

**Figure 7.** (**a**) The biomass $OD_{600nm}$ versus different amino acids in 312 h fermentation; (**b**) The hydrolase activity versus different amino acids in 312 h fermentation.

As shown in Figure 7b, the addition of amino acids had a promoting effect on the activity of hydrolytic enzymes. The best-performing group was the composite amino acid group, with a hydrolytic enzyme activity of approximately 26,800 U/mL after 312 h of fermentation, an increase of 32% compared to the control group, and the enzyme activity maintained an upward trend throughout the fermentation process. The 0.1 mol/L histidine group and the composite amino acid group showed the same upward trend in enzyme activity. However, due to the poor fluidity of their fermentation broth, it was difficult to collect and process. The hydrolytic enzyme activity of the 0.1 mol/L aspartic acid group also increased, but after 264 h of fermentation, the enzyme activity tended to stabilize with little room for increase.

### 3.2.4. Multibatch Stable Fermentation

As shown in Figure 8, after three batches of 300 h of fermentation, the biomass $OD_{600nm}$ remained stable at 454, and the hydrolytic enzyme activity remained stable at 25,000 U/mL, achieving efficient expression of lipase in a 5 L fermentation tank. Compared with the previous lipase fermentation in the laboratory [29,30], the lipase activity has improved, increasing from 9600 U/mL to 25,000 U/mL.

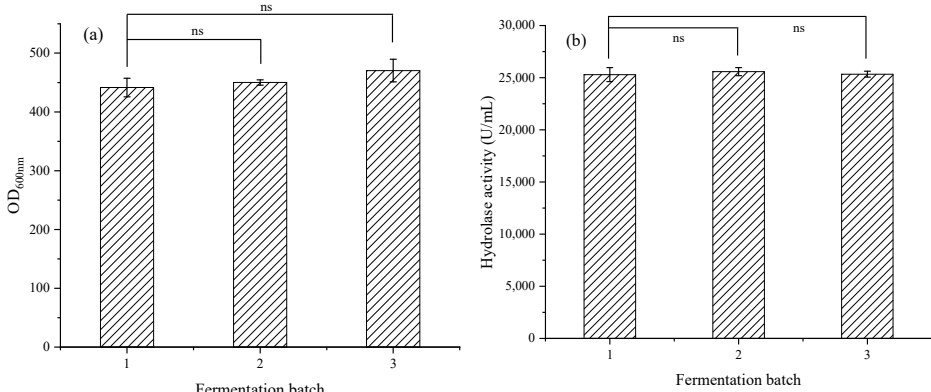

**Figure 8.** (**a**) The biomass in multibatch stable fermentation for 300 h; (**b**) The hydrolase activity in multibatch stable fermentation for 300 h. The data represent the mean $\pm$ standard deviation (*n* = 3). Statistical analysis was carried out by using the Student's *t* test (two-tailed, ns, not significant).

### 3.3. Optimization of Production Process for Free Fatty Acids

3.3.1. Optimization of Material Flow Rate in Free Fatty Acids Production Process

As shown in Figure 9a, at the first hour of the reaction, the reaction rate increased rapidly, and the yield of free fatty acids increased with the increase of the material flow rate. When the flow rate was very small and lower than 1688 mL/min, the material temperature could not reach the corresponding temperature due to the slow heat exchange between the pipe wall and the material, so the minimum flow rate of the selected peristaltic pump was 1688 mL/min. At this time, the 1 h yield of free fatty acids was 34.24%. When the flow rate reached 3938 mL/min at the maximum, the yield of free fatty acids could reach 45.32% in 1 h, mainly due to the limitation of mass transfer between the oil and water phases at the beginning of the reaction. The higher the flow rate, the stronger the mass transfer effect, and the more uniform the mixing between materials, resulting in a higher yield of free fatty acids. After 1 h of reaction, the overall reaction rate decreased, mainly due to the reaction kinetics of lipase hydrolysis of soybean oil. Figure 10 shows that the yield of free fatty acids reached its maximum at a flow rate of 3375 mL/min, reaching 94.59% after 12 h of reaction. A further increase in the flow rate resulted in a decrease in the yield of free fatty acids. This was mainly because an excessive flow rate would have a greater shear effect on the lipase, leading to its deactivation. At the flow rate of 3375 mL/min, although the maximum yield of free fatty acids could be reached, the product that just came out was colored, indicating that there was a certain amount of iron from the 3D-printing static reactor which might cause some damage to the lipase. Considering the reuse of lipase, the material flow rate of 2813 mL/min was used to further explore the next process parameters.

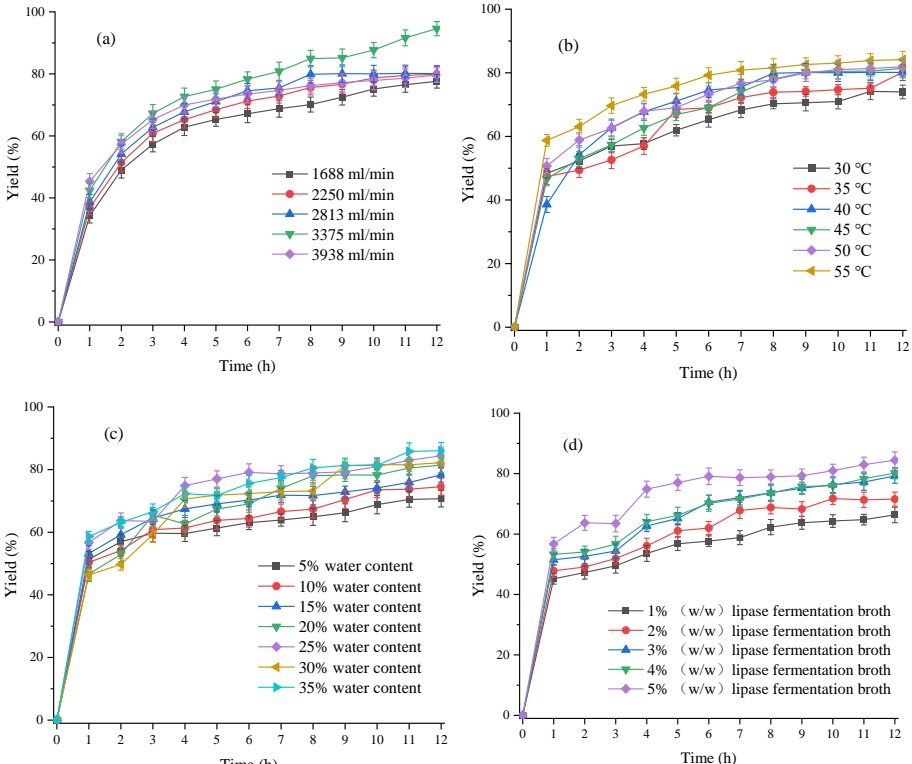

**Figure 9.** (**a**) Effect of material flow rate on the yield of free fatty acids. Other reaction conditions: reaction temperature 40 °C, water content 20 wt% and lipase fermentation broth consumption 5 wt%; (**b**) Effect of reaction temperature on the yield of free fatty acids. Other reaction conditions: material flow rate 2813 mL/min, water content 20 wt% and lipase fermentation broth consumption 5 wt%; (**c**) Effect of water content on the yield of free fatty acids. Other reaction conditions: material flow rate 2813 mL/min, reaction temperature 45 °C and lipase fermentation broth consumption 5 wt%; (**d**) Effect of lipase dosage on the yield of free fatty acids. Other reaction conditions: material flow rate 2813 mL/min, reaction temperature 45 °C and water content 25% (*w/w*).

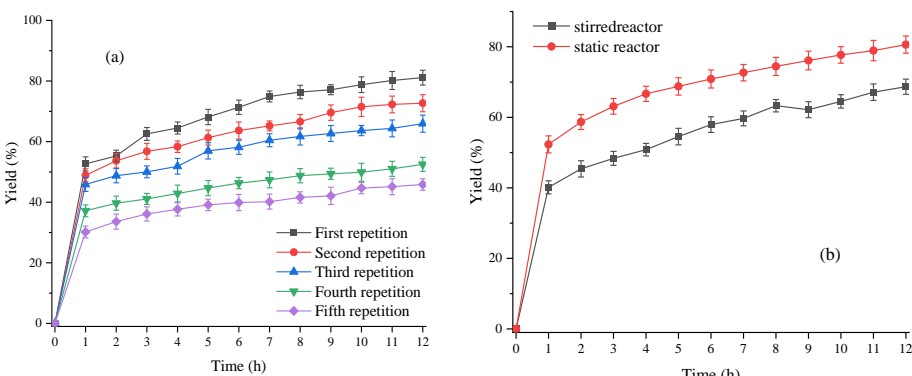

**Figure 10.** (**a**) Reuse of lipase fermentation broth. Reaction conditions: material flow rate 2813 mL/min, reaction temperature 45 °C, water content 25 wt% and lipase consumption 3 wt%; (**b**) Comparison of experimental data on the production of free fatty acids in a self-made static reactor and in a stirred reactor.

### 3.3.2. Optimization of Reaction Temperature in the Production Process of Free Fatty Acids

As shown in Figure 9b, overall, the higher the temperature, the higher the yield of free fatty acids. After 12 h of reaction, at a reaction temperature of 30 °C, the yield of free fatty acids was 73.98%, while at a reaction temperature of 55 °C, the yield of free fatty acids could reach 84.14%. Considering that the yield of free fatty acids was 81.41% at a temperature of 45 °C, which was not significantly different from the yield of free fatty acids at a temperature of 55 °C, therefore, the reaction temperature 45 °C would be applied to the next step of exploring the conditions of process parameters—water content taking into account the yield of free fatty acids and the impact of production energy consumption.

### 3.3.3. Optimization of Water Content in the Production Process of Free Fatty Acids

As shown in Figure 9c, the overall yield of free fatty acids increased with the increase of water content. When the water content was only 5 wt%, the yield of free fatty acids was relatively low, only 70.71%. When the water content increased to 35 wt%, the yield of free fatty acids could reach 86.04%. Furthermore, increasing the water content further resulted in the collected sample presenting as a semisolid emulsion liquid, centrifuged at 12,000 rpm for 3 min, and the oil–water phase that formed a stable emulsion could not be separated. When the water content was 25 wt%, the yield of free fatty acids could reach 84.42%, while increasing the water content by 10 wt% only increased the yield of free fatty acids by less than 2%. Therefore, a 25 wt% water content is used for the next step of research.

### 3.3.4. Optimization of Lipase Fermentation Broth Dosage in the Production Process of Free Fatty Acids

As shown in Figure 9d, the yield of free fatty acids significantly increased with the increase of lipase dosage. The larger the lipase dosage, the more active sites it provided and the faster the hydrolysis rate would be. When the lipase dosage was only 1 wt%, the yield of free fatty acids was only 66.33%. When the lipase dosage was increased to 5 wt%, the yield of free fatty acids could reach 84.42%, increased by approximately 18%. Considering the cost issue of lipase, when the lipase dosage was only 3 wt%, the yield of free fatty acids was approximately 80%. Therefore, the lipase dosage of 3% (*w*/*w*) is used when investigating the reuse of enzymes in the later stage.

The current optimal process for producing free fatty acids from soybean oil by lipase hydrolysis was as follows: material flow rate 2813 mL/min, reaction temperature 45 °C, water content 25 wt% and lipase consumption 3 wt%. Under optimal conditions, the yield of free fatty acids was 81.13%.

### 3.3.5. Reuse of Lipase Fermentation Broth

As shown in Figure 10a, the lipase fermentation aqueous solution was reused three times and still maintained 81.25% enzyme activity. At this time, the fatty acid yield was 65.92%. However, Nie et al. [31,32] could use this lipase to produce biodiesel five or six times, and the activity of the lipase was still above 80% due to adding cyclodextrin or glucose as the protective agent of lipase, which had a stabilizing effect on the three-dimensional structure of lipase, and due to the static mixer having a larger shear force, compared with the stirring paddle, which had a destructive effect on the three-dimensional structure of lipase.

### 3.3.6. Comparison of Fatty Acid Production in Stirred Reactors and Static Mixers

Having optimized the process of lipase hydrolysis of soybean oil to produce free fatty acids in a self-made static reactor, and having compared the process in this device with that in a stirred reactor, the selected process of lipase hydrolysis of soybean oil to produce free fatty acids in a stirred reactor was as follows: reaction temperature 45 °C, stirring speed 600 rpm, lipase fermentation broth 3 wt%, and water content 25 wt%. The final comparison results are shown in Figure 10b.

From Figure 10b, it can be seen that the yield of free fatty acids in the stirred tank reactor for 1 h was only 40.15%, while the yield of free fatty acids in the self-made static reactor for 1 h reached 52.32%. This indicated that the self-made static reactor had an enhanced effect on the mass transfer of the oil and water phases. In addition, the yield of free fatty acids reached 80.63% after 12 h in the static reactor (The GC data of the reaction procedure under optimum conditions was shown in Table S2), while it was only 68.68% in the stirred reactor. Therefore, the self-made static reactor had a strengthening effect on the process of lipase hydrolysis of soybean oil into free fatty acids, which was consistent with the results reported in previous laboratory studies on this self-made static reactor [23].

## 4. Limitation and Discussion

Through the above research, a high fermenting enzyme activity lipase with industrial potential of 25,000 U/mL was obtained, which was much higher than the 9600 U/mL reported in the previous laboratory literature [30], demonstrating the enormous industrial value of this lipase fermentation process. However, this study took 300 h of fermentation time, and further shortening of fermentation time was needed to save energy consumption. Moreover, the fermentation was carried out in a 5 L fermentation tank, which was still a certain gap from industrial production. It was necessary to conduct industrial scale-up validation experiments at a 1-ton fermentation tank in further research. In addition, this study optimized the enzymatic production process of liquid biofuel precursor—free fatty acids. After 12 h of reaction in a self-made static mixed reactor, a yield of 81.13% was obtained, which was 11.95% higher than that in a traditional stirred reactor, showcasing strong industrial potential. However, the longer reaction time limited the increase in its yield. In the later stage, the reaction time could be shortened by combining enzymes or adding trace amounts of alkali.

## 5. Conclusions

Firstly, the growth curve of the *Y. lipolytica* strains L11-01 with high lipase production was determined, and the shaking flask fermentation cycle was determined to be 120 h. The carbon and nitrogen sources in the flask culture medium were optimized. A quantity of 3 g/L citric acid could promote the expression of lipase, while high concentrations of citric acid could cause strains' death and increase the activity of hydrolytic enzymes to 1025 U/mL. After screening among 20 amino acids, it was found that histidine, glycine and alanine had a promoting effect on biomass; tyrosine, lysine, aspartic acid and histidine had a promoting effect on the activity of hydrolases. On this basis, the effects of different concentrations of amino acids on strains' growth and enzyme production were studied.

Secondly, based on the above optimization of shaking flask experiments, further optimization of carbon and nitrogen sources was carried out in a 5 L fermentation tank, and the shaking flask experiment results were validated. Adding citric acid six times and adding 5 g/L resulted in a hydrolytic enzyme activity of 13,625 U/mL, which increased by 40% compared to the control group. However, the biomass decreased by half compared to the control group, which could be mutually verified with the results of shaking flask fermentation. By measuring the consumption of soybean oil during the fermentation process, a carbon source automatic feeding scheme was designed: 0–96 h feeding rate 8.3 g/h; 96–240 h feeding rate 6.25 g/h; 240–312 h feeding rate 4.2 g/h. After adjusting the feeding method, the biomass $OD_{600nm}$ was increased to 530, and the hydrolytic enzyme activity was increased to 20,220 U/mL. On the basis of optimizing the shaking flask composite amino acids, the addition of single and composite amino acids during the amplification fermentation process achieved an increase in lipase expression. By adding composite amino acids, the final hydrolytic enzyme activity was further increased to 26,800 U/mL, and stabilized at 25,000 U/mL after 300 h of fermentation. The combination of three optimization strategies achieved a significant increase in hydrolytic enzyme activity, achieving efficient expression of lipase in a 5 L fermentation tank.

Finally, on the basis of optimizing the culture conditions of *Y. lipolytica* and regulating the efficient expression of lipase in a 5 L fermentation tank, a lipase fermentation broth with a laboratory hydrolytic enzyme activity of 25,000 U/mL was applied to a self-made static reactor in the laboratory to hydrolyze soybean oil into free fatty acids. The optimal reaction conditions were as follows: flow rate of material 2813 mL/min, reaction temperature 45 °C, water content 3 wt% and lipase fermentation broth 3 wt%. Under the optimal reaction conditions, the yield of free fatty acids downstream was 80.63%. Lipase aqueous solution could be reused three times and enzyme activity maintained above 80%. At the same time, compared with the stirred reactor, the self-made static reactor has enhanced the mass transfer of oil–water phases.

**Supplementary Materials:** The following supporting information can be downloaded at: https://www.mdpi.com/article/10.3390/fermentation9080708/s1, Figure S1: Sepax AAA detection chromatogram; Figure S2: Determination of the growth curve of the selected lipase strains to determine the subsequent experimental days.; Table S1: Mobile phase gradient; Table S2: GC data of the reaction procedure under optimum conditions.

**Author Contributions:** S.H.: writing—original draft and investigation. H.L.: partial data curation. R.Z.: partial data curation. M.W.: conceptualization and supervision. T.T.: funding acquisition. All authors have read and agreed to the published version of the manuscript.

**Funding:** This research was funded by the National Key R&D Program of China, grant number 2021YFC2103702.

**Data Availability Statement:** Data is not available due to privacy.

**Conflicts of Interest:** The authors declare no conflict of interest.

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
