# Peer review of "Production of Liquid Biofuel Precursors: Optimization and Regulation of Lipase Fermentation and Its Application in Plant Oil Hydrolysis Process"

_fermentation, doi:10.3390/fermentation9080708_

Round 1

Reviewer 1 Report

Title: Production of liquid biofuel precursors: Optimization and regulation of lipase fermentation and its application in plant oil hydrolysis process

Overall comments

The paper describes the optimization and regulation of lipase fermentation for producing liquid biofuel precursors, explicitly focusing on the hydrolysis of soybean oil. The paper contains comprehensive information, and the authors have reported their findings effectively. After major revisions, this paper can be accepted for publication.

Comments on the title, Abstract, and References

The title is ok for the study.

The abstract provides information about the reaction conditions and results. However, it should also highlight the novelty of the research and its potential applications. Additionally, the authors should include statistical analysis results. Authors should include more updated references if available. 

Comments on Introduction

The introduction is informative, but some parts require revision with proper referencing.

"Currently, fossil fuels are still the world's main energy source, accounting for approximately 35% of the total energy output." Reference 1 does not support this statement. Please verify and provide up-to-date references, if available.

"At present, there are three main ways to produce free fatty acids: steam cracking, saponification, and enzymatic hydrolysis." Please include relevant references to support this statement.

Line 82-90: Please provide appropriate references to support the mentioned information.

Comments on Experiments

"Column temperature rises from 100℃/min to 180℃ at 15℃/min, held for 5 minutes, and then rises from 180℃ to 350℃ at 20℃/min, held for 15 minutes." Please clarify the intended meaning of this statement.

Ammonia and phosphoric acid are missing in the Materials section.

What do the authors mean by "100% ammonia"?

"Ammonia is automatically added to maintain pH 4.8." Please provide more details on the process of automated pH control using ammonia.

"8% (v/v) soybean oil." Please clarify the meaning of this statement.

Line 187: "a certain temperature." What was the specific temperature used? "a certain amount of soybean." What was the quantity used? "a certain amount of lipase." What was the quantity used?

The experimental optimization process should be included in this section, and the results should be discussed in the Results and Discussion section.

The resolution of each figure needs to be improved.

Comments on Results and Discussion

Line 263-264: "After screening, it was found that histidine had a good promoting effect on both." Please provide the percentage of improvement compared to the control.

Fig. 14: Why does the curve of the static reactor show a zigzag motion? As a reviewer, I am providing this comment for the authors' revision. Please correct any language-related issues.

Conclusions

It short is short and informative. It should focus on the conclusion of this study with limitations, future research perspectives, and utility. Authors may include a discussion and a limitation section before the conclusion. 

Reviewer 2 Report

I hope the following information will be helpful in deciding the fate of the article presented : fermentation - 2501781

 This publication addresses the very important issue of process optimisation towards stable, reproducible, and optimal conditions so that lipase can be efficiently applied in the enzymatic hydrolysis of soybean oil to obtain free fatty acids.

The work and the results of the research carried out have been presented in a thoughtful and understandable manner. I believe that the material presented is of great value, as it presents important findings and explores various options that have had a significant impact on the optimisation of the enzymatic hydrolysis process at different scales.

In my opinion, the article "Production of liquid biofuel precursors: Optimisation and regulation of lipase fermentation and its application in plant oil hydrolysis process" is of high scientific value, but requires necessary additions.

1) In pp. 2.2, it should be explained that "The content of free fatty acids (FFAs) is determined by the relative content of the substance"???? The methodology under this point should be described in more detail.

2) In developing the chromatographic method, the authors must have used some kind of standard mixture of "free fatty acids". This information should be included in section 2.1. 2.1 Materials

3) It is certain that the qualitative content of "free fatty acids" varied considerably during the research work. Such studies should, for example, be supplemented for optimal conditions.

4) The summary does not include information on the yield of free fatty acids under optimal conditions using the static reactor developed.

Reviewer 3 Report

the article is well written and only conclusion should be improved. It should be more clear the novelty and impact on the society. The discussion section and also comparison with the literature should be added where all results are compared to what has been previously done. The limitation of the current findings should be also more visible and integrated into the discussion.

Round 2

Reviewer 1 Report

Dear Authors,

Thank you for your good job on the revised manuscript. I am pleased to recommend your manuscript for publication, pending minor revisions. Please find below the comments and suggestions for improvement:

  1. Units: In the manuscript text and figures, I observed the use of different units, such as "ml" and "mL" or "min" and "minute," as well as "hours" and "h." Kindly ensure consistency by revising the units throughout the manuscript.
  2. Figure Resolution: To enhance the clarity of the figures, especially in terms of error bars on the curves (e.g., Fig. 2, 4, 6, 7, 8, 9), please increase the resolution or line width accordingly.
  3. Figure Arrangement: It is advisable to combine Figures 9, 10, 11, and 12 as one group and Figures 13 and 14 as another for better organization and readability.
  4. Figure Notations: To improve the figures' readability, please label each subfigure as (a), (b), (c), (d), etc., and ensure that the corresponding captions reflect the correct notations.
  5. Figure 3: In Figure 3, the notations of (a), (b), (c), and (d) are missing. Additionally, the caption for subfigures (e), (f), (g), and (h) needs to be added for clarity.
  6. Captions for Figures 4, 6, 7, and 8: Please review the captions for these figures and appropriately mark subfigures as (a) and (b) for consistency.
  7. Manuscript Template: To ensure uniformity in the submission process, I suggest using a Microsoft Word template provided by the journal for the manuscript.
